# Texture Synthesis Using Convolutional Neural Networks

**Leon A. Gatys**
Centre for Integrative Neuroscience, University of Tübingen, Germany
Bernstein Center for Computational Neuroscience, Tübingen, Germany
Graduate School of Neural Information Processing, University of Tübingen, Germany
`leon.gatys@bethgelab.org`

**Alexander S. Ecker**
Centre for Integrative Neuroscience, University of Tübingen, Germany
Bernstein Center for Computational Neuroscience, Tübingen, Germany
Max Planck Institute for Biological Cybernetics, Tübingen, Germany
Baylor College of Medicine, Houston, TX, USA

**Matthias Bethge**
Centre for Integrative Neuroscience, University of Tübingen, Germany
Bernstein Center for Computational Neuroscience, Tübingen, Germany
Max Planck Institute for Biological Cybernetics, Tübingen, Germany

## Abstract

Here we introduce a new model of natural textures based on the feature spaces of convolutional neural networks optimised for object recognition. Samples from the model are of high perceptual quality demonstrating the generative power of neural networks trained in a purely discriminative fashion. Within the model, textures are represented by the correlations between feature maps in several layers of the network. We show that across layers the texture representations increasingly capture the statistical properties of natural images while making object information more and more explicit. The model provides a new tool to generate stimuli for neuroscience and might offer insights into the deep representations learned by convolutional neural networks.

## 1 Introduction

The goal of visual texture synthesis is to infer a generating process from an example texture, which then allows to produce arbitrarily many new samples of that texture. The evaluation criterion for the quality of the synthesised texture is usually human inspection and textures are successfully synthesised if a human observer cannot tell the original texture from a synthesised one.

In general, there are two main approaches to find a texture generating process. The first approach is to generate a new texture by resampling either pixels [5, 28] or whole patches [6, 16] of the original texture. These non-parametric resampling techniques and their numerous extensions and improvements (see [27] for review) are capable of producing high quality natural textures very efficiently. However, they do not define an actual model for natural textures but rather give a mechanistic procedure for how one can randomise a source texture without changing its perceptual properties.

In contrast, the second approach to texture synthesis is to explicitly define a parametric texture model. The model usually consists of a set of statistical measurements that are taken over the

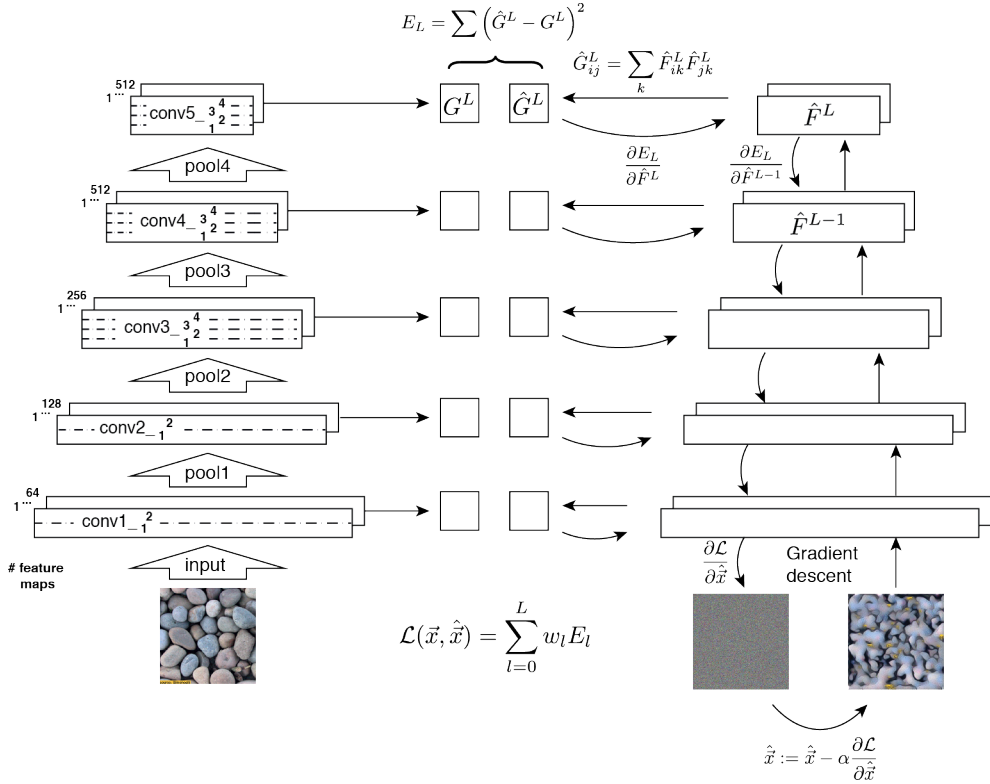

$$E_L = \sum \left( \hat{G}^L - G^L \right)^2$$

$$\hat{G}^L_{ij} = \sum_k \hat{F}^L_{ik} \hat{F}^L_{jk}$$

$$\mathcal{L}(\vec{x}, \hat{\vec{x}}) = \sum_{l=0}^{L} w_l E_l$$

$$\hat{\vec{x}} := \hat{\vec{x}} - \alpha \frac{\partial \mathcal{L}}{\partial \hat{\vec{x}}}$$

Figure 1: Synthesis method. Texture analysis (left). The original texture is passed through the CNN and the Gram matrices $G_l$ on the feature responses of a number of layers are computed. Texture synthesis (right). A white noise image $\hat{\vec{x}}$ is passed through the CNN and a loss function $E_l$ is computed on every layer included in the texture model. The total loss function $\mathcal{L}$ is a weighted sum of the contributions $E_l$ from each layer. Using gradient descent on the total loss with respect to the pixel values, a new image is found that produces the same Gram matrices $\hat{G}_l$ as the original texture.

spatial extent of the image. In the model a texture is uniquely defined by the outcome of those measurements and every image that produces the same outcome should be perceived as the same texture. Therefore new samples of a texture can be generated by finding an image that produces the same measurement outcomes as the original texture. Conceptually this idea was first proposed by Julesz [13] who conjectured that a visual texture can be uniquely described by the Nth-order joint histograms of its pixels. Later on, texture models were inspired by the linear response properties of the mammalian early visual system, which resemble those of oriented band-pass (Gabor) filters [10, 21]. These texture models are based on statistical measurements taken on the filter responses rather than directly on the image pixels. So far the best parametric model for texture synthesis is probably that proposed by Portilla and Simoncelli [21], which is based on a set of carefully handcrafted summary statistics computed on the responses of a linear filter bank called *Steerable Pyramid* [24]. However, although their model shows very good performance in synthesising a wide range of textures, it still fails to capture the full scope of natural textures.

In this work, we propose a new parametric texture model to tackle this problem (Fig. 1). Instead of describing textures on the basis of a model for the early visual system [21, 10], we use a convolutional neural network – a functional model for the entire ventral stream – as the foundation for our texture model. We combine the conceptual framework of spatial summary statistics on feature responses with the powerful feature space of a convolutional neural network that has been trained on object recognition. In that way we obtain a texture model that is parameterised by spatially invariant representations built on the hierarchical processing architecture of the convolutional neural network.

## 2 Convolutional neural network

We use the VGG-19 network, a convolutional neural network trained on object recognition that was introduced and extensively described previously [25]. Here we give only a brief summary of its architecture.

We used the feature space provided by the 16 convolutional and 5 pooling layers of the VGG-19 network. We did not use any of the fully connected layers. The network's architecture is based on two fundamental computations:

1. Linearly rectified convolution with filters of size $3 \times 3 \times k$ where $k$ is the number of input feature maps. Stride and padding of the convolution is equal to one such that the output feature map has the same spatial dimensions as the input feature maps.

2. Maximum pooling in non-overlapping $2 \times 2$ regions, which down-samples the feature maps by a factor of two.

These two computations are applied in an alternating manner (see Fig. 1). A number of convolutional layers is followed by a max-pooling layer. After each of the first three pooling layers the number of feature maps is doubled. Together with the spatial down-sampling, this transformation results in a reduction of the total number of feature responses by a factor of two. Fig. 1 provides a schematic overview over the network architecture and the number of feature maps in each layer. Since we use only the convolutional layers, the input images can be arbitrarily large. The first convolutional layer has the same size as the image and for the following layers the ratio between the feature map sizes remains fixed. Generally each layer in the network defines a non-linear filter bank, whose complexity increases with the position of the layer in the network.

The trained convolutional network is publicly available and its usability for new applications is supported by the caffe-framework [12]. For texture generation we found that replacing the max-pooling operation by average pooling improved the gradient flow and one obtains slightly cleaner results, which is why the images shown below were generated with average pooling. Finally, for practical reasons, we rescaled the weights in the network such that the mean activation of each filter over images and positions is equal to one. Such re-scaling can always be done without changing the output of a neural network as long as the network is fully piece-wise linear [1].

## 3 Texture model

The texture model we describe in the following is much in the spirit of that proposed by Portilla and Simoncelli [21]. To generate a texture from a given source image, we first extract features of different sizes homogeneously from this image. Next we compute a spatial summary statistic on the feature responses to obtain a stationary description of the source image (Fig. 1A). Finally we find a new image with the same stationary description by performing gradient descent on a random image that has been initialised with white noise (Fig. 1B).

The main difference to Portilla and Simoncelli's work is that instead of using a linear filter bank and a set of carefully chosen summary statistics, we use the feature space provided by a high-performing deep neural network and only one spatial summary statistic: the correlations between feature responses in each layer of the network.

To characterise a given vectorised texture $\vec{x}$ in our model, we first pass $\vec{x}$ through the convolutional neural network and compute the activations for each layer $l$ in the network. Since each layer in the network can be understood as a non-linear filter bank, its activations in response to an image form a set of filtered images (so-called *feature maps*). A layer with $N_l$ distinct filters has $N_l$ feature maps each of size $M_l$ when vectorised. These feature maps can be stored in a matrix $F^l \in \mathcal{R}^{N_l \times M_l}$, where $F_{jk}^l$ is the activation of the $j^{\text{th}}$ filter at position $k$ in layer $l$. Textures are per definition stationary, so a texture model needs to be agnostic to spatial information. A summary statistic that discards the spatial information in the feature maps is given by the correlations between the responses of

different features. These feature correlations are, up to a constant of proportionality, given by the Gram matrix $G^l \in \mathcal{R}^{N_l \times N_l}$, where $G^l_{ij}$ is the inner product between feature map $i$ and $j$ in layer $l$:

$$G^l_{ij} = \sum_k F^l_{ik} F^l_{jk}. \tag{1}$$

A set of Gram matrices $\{G^1, G^2, ..., G^L\}$ from some layers $1, \dots, L$ in the network in response to a given texture provides a stationary description of the texture, which fully specifies a texture in our model (Fig. 1A).

## 4 Texture generation

To generate a new texture on the basis of a given image, we use gradient descent from a white noise image to find another image that matches the Gram-matrix representation of the original image. This optimisation is done by minimising the mean-squared distance between the entries of the Gram matrix of the original image and the Gram matrix of the image being generated (Fig. 1B).

Let $\vec{x}$ and $\hat{\vec{x}}$ be the original image and the image that is generated, and $G^l$ and $\hat{G}^l$ their respective Gram-matrix representations in layer $l$ (Eq. 1). The contribution of layer $l$ to the total loss is then

$$E_l = \frac{1}{4N_l^2 M_l^2} \sum_{i,j} \left( G^l_{ij} - \hat{G}^l_{ij} \right)^2 \tag{2}$$

and the total loss is

$$\mathcal{L}(\vec{x}, \hat{\vec{x}}) = \sum_{l=0}^{L} w_l E_l \tag{3}$$

where $w_l$ are weighting factors of the contribution of each layer to the total loss. The derivative of $E_l$ with respect to the activations in layer $l$ can be computed analytically:

$$\frac{\partial E_l}{\partial \hat{F}^l_{ij}} = \begin{cases} \frac{1}{N_l^2 M_l^2} \left( (\hat{F}^l)^{\mathrm{T}} \left( G^l - \hat{G}^l \right) \right)_{ji} & \text{if } \hat{F}^l_{ij} > 0 \\ 0 & \text{if } \hat{F}^l_{ij} < 0 \,. \end{cases} \tag{4}$$

The gradients of $E_l$, and thus the gradient of $\mathcal{L}(\vec{x}, \hat{\vec{x}})$, with respect to the pixels $\hat{\vec{x}}$ can be readily computed using standard error back-propagation [18]. The gradient $\frac{\partial \mathcal{L}}{\partial \hat{\vec{x}}}$ can be used as input for some numerical optimisation strategy. In our work we use L-BFGS [30], which seemed a reasonable choice for the high-dimensional optimisation problem at hand. The entire procedure relies mainly on the standard forward-backward pass that is used to train the convolutional network. Therefore, in spite of the large complexity of the model, texture generation can be done in reasonable time using GPUs and performance-optimised toolboxes for training deep neural networks [12].

## 5 Results

We show textures generated by our model from four different source images (Fig. 2). Each row of images was generated using an increasing number of layers in the texture model to constrain the gradient descent (the labels in the figure indicate the top-most layer included). In other words, for the loss terms above a certain layer we set the weights $w_l = 0$, while for the loss terms below and including that layer, we set $w_l = 1$. For example the images in the first row ('conv1_1') were generated only from the texture representation of the first layer ('conv1_1') of the VGG network. The images in the second row ('pool1') where generated by jointly matching the texture representations on top of layer 'conv1_1', 'conv1_2' and 'pool1'. In this way we obtain textures that show what structure of natural textures are captured by certain computational processing stages of the texture model.

The first three columns show images generated from natural textures. We find that constraining all layers up to layer 'pool4' generates complex natural textures that are almost indistinguishable from the original texture (Fig. 2, fifth row). In contrast, when constraining only the feature correlations on the lowest layer, the textures contain little structure and are not far from spectrally matched noise

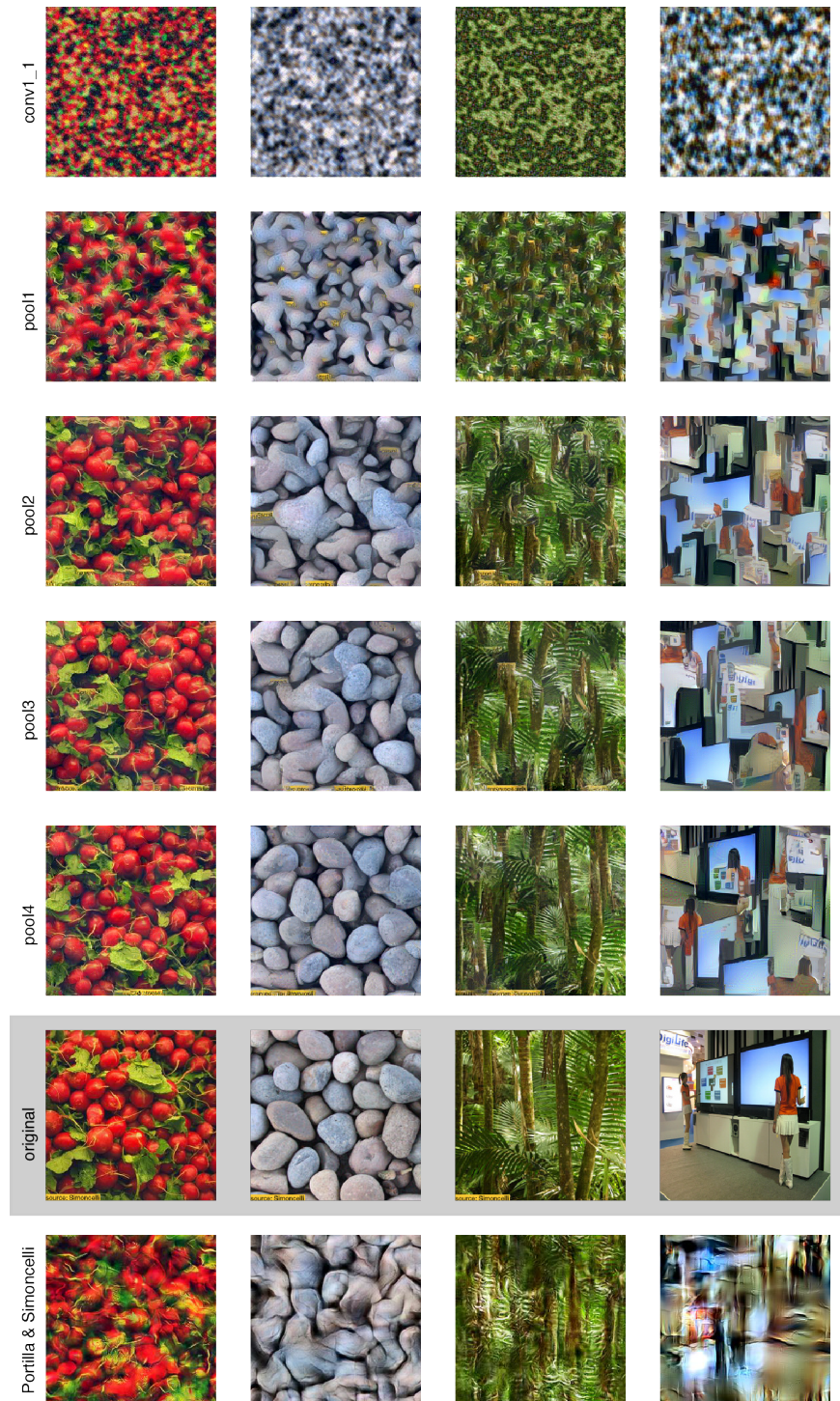

Figure 2: Generated stimuli. Each row corresponds to a different processing stage in the network. When only constraining the texture representation on the lowest layer, the synthesised textures have little structure, similarly to spectrally matched noise (first row). With increasing number of layers on which we match the texture representation we find that we generate images with increasing degree of naturalness (rows 2–5; labels on the left indicate the top-most layer included). The source textures in the first three columns were previously used by Portilla and Simoncelli [21]. For better comparison we also show their results (last row). The last column shows textures generated from a non-texture image to give a better intuition about how the texture model represents image information.

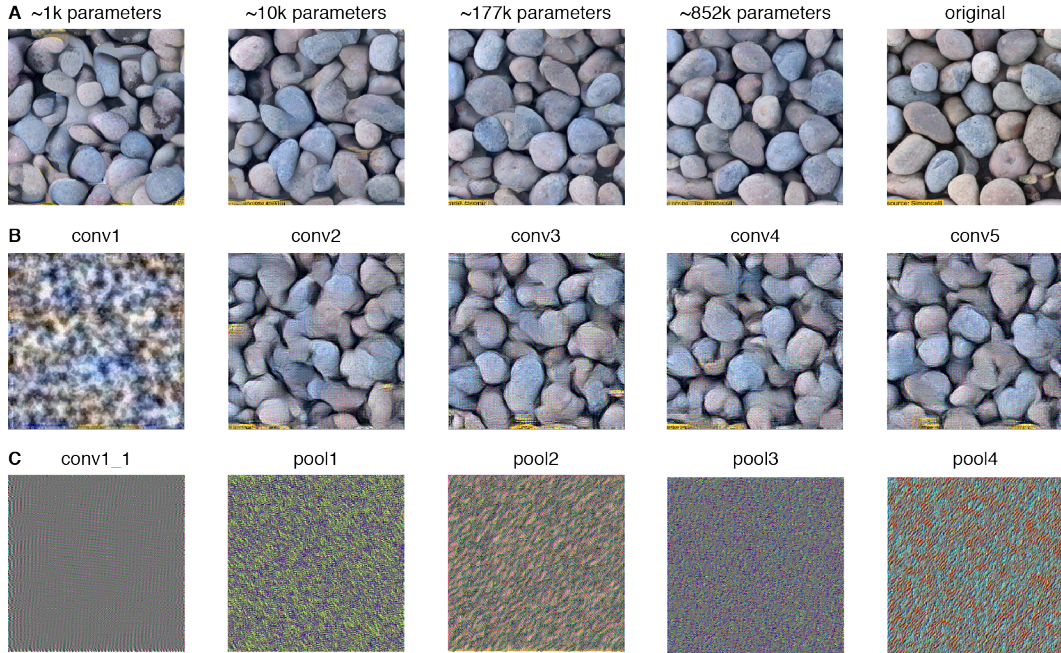

Figure 3: **A**, Number of parameters in the texture model. We explore several ways to reduce the number of parameters in the texture model (see main text) and compare the results. **B**, Textures generated from the different layers of the caffe reference network [12, 15]. The textures are of lesser quality than those generated with the VGG network. **C**, Textures generated with the VGG architecture but random weights. Texture synthesis fails in this case, indicating that learned filters are crucial for texture generation.

(Fig. 2, first row). We can interpolate between these two extremes by using only the constraints from all layers up to some intermediate layer. We find that the statistical structure of natural images is matched on an increasing scale as the number of layers we use for texture generation increases. We did not include any layers above layer 'pool4' since this did not improve the quality of the synthesised textures. For comparability we used source textures that were previously used by Portilla and Simoncelli [21] and also show the results of their texture model (Fig. 2, last row). [2]

To give a better intuition for how the texture synthesis works, we also show textures generated from a non-texture image taken from the ImageNet validation set [23] (Fig. 2, last column). Our algorithm produces a texturised version of the image that preserves local spatial information but discards the global spatial arrangement of the image. The size of the regions in which spatial information is preserved increases with the number of layers used for texture generation. This property can be explained by the increasing receptive field sizes of the units over the layers of the deep convolutional neural network.

When using summary statistics from all layers of the convolutional neural network, the number of parameters of the model is very large. For each layer with $N_l$ feature maps, we match $N_l \times (N_l + 1)/2$ parameters, so if we use all layers up to and including 'pool4', our model has $\sim 852$k parameters (Fig. 3A, fourth column). However, we find that this texture model is heavily over-parameterised. In fact, when using only one layer on each scale in the network (i.e. 'conv1_1',

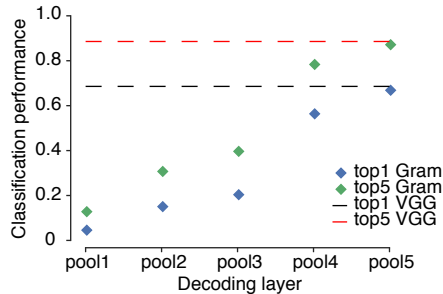

Figure 4: Performance of a linear classifier on top of the texture representations in different layers in classifying objects from the ImageNet dataset. High-level information is made increasingly explicit along the hierarchy of our texture model.

and 'pool1-4'), the model contains $\sim 177$k parameters while hardly loosing any quality (Fig. 3A, third column). We can further reduce the number of parameters by doing PCA of the feature vector in the different layers of the network and then constructing the Gram matrix only for the first $k$ principal components. By using the first 64 principal components for layers 'conv1_1', and 'pool1-4' we can further reduce the model to $\sim 10$k parameters (Fig. 3A, second column). Interestingly, constraining only the feature map averages in layers 'conv1_1', and 'pool1-4', (1024 parameters), already produces interesting textures (Fig. 3A, first column). These *ad hoc* methods for parameter reduction show that the texture representation can be compressed greatly with little effect on the perceptual quality of the synthesised textures. Finding minimal set of parameters that reproduces the quality of the full model is an interesting topic of ongoing research and beyond the scope of the present paper. A larger number of natural textures synthesised with the $\approx 177$k parameter model can be found in the Supplementary Material as well as on our website[3]. There one can also observe some failures of the model in case of very regular, man-made structures (e.g. brick walls).

In general, we find that the very deep architecture of the VGG network with small convolutional filters seems to be particularly well suited for texture generation purposes. When performing the same experiment with the caffe reference network [12], which is very similar to the AlexNet [15], the quality of the generated textures decreases in two ways. First, the statistical structure of the source texture is not fully matched even when using all constraints (Fig 3B, 'conv5'). Second, we observe an artifactual grid that overlays the generated textures (Fig 3B). We believe that the artifactual grid originates from the larger receptive field sizes and strides in the caffe reference network.

While the results from the caffe reference network show that the architecture of the network is important, the learned feature spaces are equally crucial for texture generation. When synthesising a texture with a network with the VGG architecture but random weights, texture generation fails (Fig. 3C), underscoring the importance of using a trained network.

To understand our texture features better in the context of the original object recognition task of the network, we evaluated how well object identity can be linearly decoded from the texture features in different layers of the network. For each layer we computed the Gram-matrix representation of each image in the ImageNet training set [23] and trained a linear soft-max classifier to predict object identity. As we were not interested in optimising prediction performance, we did not use any data augmentation and trained and tested only on the $224 \times 224$ centre crop of the images. We computed the accuracy of these linear classifiers on the ImageNet validation set and compared them to the performance of the original VGG-19 network also evaluated on the $224 \times 224$ centre crops of the validation images.

The analysis suggests that our texture representation continuously disentangles object identity information (Fig. 4). Object identity can be decoded increasingly well over the layers. In fact, linear decoding from the final pooling layer performs almost as well as the original network, suggesting that our texture representation preserves almost all high-level information. At first sight this might appear surprising since the texture representation does not necessarily preserve the global structure of objects in non-texture images (Fig. 2, last column). However, we believe that this "inconsis-

tency" is in fact to be expected and might provide an insight into how CNNs encode object identity. The convolutional representations in the network are shift-equivariant and the network's task (object recognition) is agnostic to spatial information, thus we expect that object information can be read out independently from the spatial information in the feature maps. We show that this is indeed the case: a linear classifier on the Gram matrix of layer 'pool5' comes close to the performance of the full network (87.7% vs. 88.6% top 5 accuracy, Fig. 4).

## 6   Discussion

We introduced a new parametric texture model based on a high-performing convolutional neural network. Our texture model exceeds previous work as the quality of the textures synthesised using our model shows a substantial improvement compared to the current state of the art in parametric texture synthesis (Fig. 2, fourth row compared to last row).

While our model is capable of producing natural textures of comparable quality to non-parametric texture synthesis methods, our synthesis procedure is computationally more expensive. Nevertheless, both in industry and academia, there is currently much effort taken in order to make the evaluation of deep neural networks more efficient [11, 4, 17]. Since our texture synthesis procedure builds exactly on the same operations, any progress made in the general field of deep convolutional networks is likely to be transferable to our texture synthesis method. Thus we expect considerable improvements in the practical applicability of our texture model in the near future.

By computing the Gram matrices on feature maps, our texture model transforms the representations from the convolutional neural network into a stationary feature space. This general strategy has recently been employed to improve performance in object recognition and detection [9] or texture recognition and segmentation [3]. In particular Cimpoi et al. report impressive performance in material recognition and scene segmentation by using a stationary Fisher-Vector representation built on the highest convolutional layer of readily trained neural networks [3]. In agreement with our results, they show that performance in natural texture recognition continuously improves when using higher convolutional layers as the input to their Fisher-Vector representation. As our main aim is to synthesise textures, we have not evaluated the Gram matrix representation on texture recognition benchmarks, but would expect that it also provides a good feature space for those tasks.

In recent years, texture models inspired by biological vision have provided a fruitful new analysis tool for studying visual perception. In particular the parametric texture model proposed by Portilla and Simoncelli [21] has sparked a great number of studies in neuroscience and psychophysics [8, 7, 1, 22, 20]. Our texture model is based on deep convolutional neural networks that are the first artificial systems that rival biology in terms of difficult perceptual inference tasks such as object recognition [15, 25, 26]. At the same time, their hierarchical architecture and basic computational properties admit a fundamental similarity to real neural systems. Together with the increasing amount of evidence for the similarity of the representations in convolutional networks and those in the ventral visual pathway [29, 2, 14], these properties make them compelling candidate models for studying visual information processing in the brain. In fact, it was recently suggested that textures generated from the representations of performance-optimised convolutional networks "may therefore prove useful as stimuli in perceptual or physiological investigations" [19]. We feel that our texture model is the first step in that direction and envision it to provide an exciting new tool in the study of visual information processing in biological systems.

**Acknowledgments**

This work was funded by the German National Academic Foundation (L.A.G.), the Bernstein Center for Computational Neuroscience (FKZ 01GQ1002) and the German Excellency Initiative through the Centre for Integrative Neuroscience Tübingen (EXC307)(M.B., A.S.E, L.A.G.)

## Footnotes

[1]Source code to generate textures with CNNs as well as the rescaled VGG-19 network can be found at http://github.com/leongatys/DeepTextures

[2]A curious finding is that the yellow box, which indicates the source of the original texture, is also placed towards the bottom left corner in the textures generated by our model. As our texture model does not store any spatial information about the feature responses, the only possible explanation for such behaviour is that some features in the network explicitly encode the information at the image boundaries. This is exactly what we find when inspecting feature maps in the VGG network: Some feature maps, at least from layer 'conv3_1' onwards, only show high activations along their edges. This might originate from the zero-padding that is used for the convolutions in the VGG network and it could be interesting to investigate the effect of such padding on learning and object recognition performance.

[3]www.bethgelab.org/deeptextures

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
