[Supplementary Material]

# Supplementary Material

Here we present a larger number of natural textures synthesised using a Convolutional Neural Network as described in the main paper. The source textures are taken from the CG texture database [1] and down-sampled such that the total number of pixels equals $256^2$. This down-sampling is done to match the scale of the images on which the network was trained and to decrease computational costs. The generation of one texture took about 10 mins running on a K40 GPU.

The textures were generated by matching the correlations between feature maps in layers 'conv1_1', and 'pool1-4' of a normalised version of the 19-layer VGG-Network. The weights in the normalised network are scaled such that the mean activation of each filter over images and positions is equal to one. Such re-scaling can always be done without changing the output of a neural network as long as the network is fully piece-wise linear. The synthesis was carried out using the Berkley Vision Caffe-framework. More examples can be found on our website at www.bethgelab.org/deeptextures.

Synthesised

Source

Synthesised

Source

Synthesised

Source

Synthesised

Source

Synthesised

Source

Synthesised

Source

Synthesised

Source

Synthesised

Source

Synthesised — Source

Synthesised — Source

Synthesised — Source

Synthesised — Source

Synthesised

Source

Synthesised

Source

Synthesised

Source

Synthesised

Source

## Footnotes

[1] http://www.textures.com