[Reviews · NeurIPS 2015]

Submitted by Assigned_Reviewer_1

This paper uses a convolutional network to generate textures with the aim of using this texture generation technique to generate stimuli that elicit predictable neural responses in different parts of the visual pathway, assuming it works like the model. They match the correlations of CNN activities of an image to a synthesized image up to some layer in a CNN.

Hopefully this will lead to some fruitful experiments.

From past work, it is obvious that generating an image constrained by some summary statistics of a deep net representation of a target image will have a greater resemblance to this target if more of the deep net representation summary statistics are used. This paper does a nice job of showing that convincing textures can be generated using this technique, and that by using statistics from more layers, the process improves.

BTW, The first work along these lines was by Bergen and Heeger in the mid-1990's, not cited.

The Portilla & Simoncelli work built on that.

Beyond indicating that the process works, the paper gives very little insight into why it works. Of course part of the reason for this is that no one really understands the representations learned by deep nets, so we are building one mystery on top of another.

The observation that an untrained network fails to generate textures is too simplistic to provide insights about how the process operates.

This paper seeks to link the deep network to biological vision. The only basis for comparison is the hierarchical nature of each. However the details of the biological system are so much richer and complicated than anything in a deep net that one must really wonder if this can be taken seriously.

For example the paper neglects the concept of feedback which is a major component of the biological system.

Nevertheless, as the authors mention, there have been previous attempts to predict neural responses in the visual hierarchy using deep nets which have been fairly successful.

Following that line of thought this seems worth exploring, but to the extent it actually succeeds in predicting neural responses it begs more questions and than it answers.

Also, the actual way in which this model would be deployed to make an experimentally testable prediction is rather vague.

This statement for example leaves much to the imagination:

"Textures generated from the different layers of our texture model should lead to equal responses up to a certain processing stage an to diverging responses in the processing stages thereafter."

If one thinks this through, it is really more of a thought experiment than a practical experiment.

One would need to find many features like those at a particular level of the network, then look at their pairwise correlations in response to the synthesized vs. real textures, and then show that they are statistically similar.

After all that, what would it really tell us?

That the visual hierarchy is a deep net?

Really?

In summary, the paper provides a nice example of generating textures using CNNs, but provides no new insights into the functioning of these networks, and the application to understanding biological vision is not very compelling.

However having said all that, overall I found it worth reading because the results are still quite interesting to look at, irrespective of the neurobiological implications of the model.

Overall the work is well done, the results are intriguing, and I believe it will be of interest to NIPS.

Post-rebuttal comments:

The authors make a convincing case in their rebuttal.

It is good to know that similar results can be achieved with many fewer parameters.

These things and other technical details should definitely be emphasized and included in the revised ms.

The neuroscience aspects, while potentially intriguing, deserve fuller development in a separate paper and so I would recommend whittling down that part here to make room to elaborate the technical aspects, which many readers/viewers will want to know.
Summary: Overall an interesting paper and result, though the applications to understanding the visual system are not particularly compelling.

Submitted by Assigned_Reviewer_2

The authors demonstrate a new method for texture synthesis using the response statistics of layers in a neural network. The algorithm using moment matching (correlation, specifically) between the statistics of a randomly generated image and a texture image that they are trying to generate. The authors are able to show texture interpolation, from spectrum matching to complex image features, by adjusting the number of layers used in the constraints.

The paper is mostly well written and easy to follow. It would be nice if figure 1 was more descriptive of the model and less provocative of the link between the model and primate visual cortex. For example, in line 124 the authors mention that, "a number of convolutional layers is followed by a max pooling layer". Which layers?

The originality of this work is hard to clarify. On the one hand, as I know this is the first work to use moment matching between a generated texture image and the layers of a neural network. This may have many future implications for understanding and generating textures. On the other hand, the authors use an out-of-the-box neural network architecture (the VGG-19) and largely replicate the work of [16], though with fewer constraints and less rigor.

The significance of this work is also difficult to judge. The paper is definitely interesting and is a thought starter. However, this seems to be its best attribute. It seems in a similar vein to the recent neural network 'hallucination' work by Mordvintsev et al from Google. In the future it may be an interesting method for understanding why neural networks are behaving as they do, but it's not clear what one learns from this paper.

A major potential red flag is that the texture generation method works so well in its full capacity. There are thousands of parameters to use when training an image to match the target texture. A trivial texture output is to just learn the value of each pixel in the input image. This may well be what is happening in this method (though without absolute positional information), as it is clear that image generation of non-textures still manage to capture salient chunks of the input image.

There are also a few questions I am left with:

(1) Why do the authors choose to use only the correlations between units at the same location within a layer? Is this why lower layers can only generate images that look like spectrum-matched noise? This could be an artificial result of the fact that each unit is only a few pixels wide and that the correlation constraint is only local. [16] found that other constraints were very useful. In fact, understanding what happens to generated images as certain constraints were lifted was a central message to their paper.

(2) Why use MSE as an objective function, besides that it's easy? Correlation matrices are bounded by -1 and 1, and the entires are not likely well described with a normal distribution.

(3) In practice, which layers are contributing most heavily to your optimization problem (when all layers are used)? What happens if you remove the constrain on lower layers and only constrain with the higher layers?

(4) How would this work on non-visual textures? (e.g. http://www.ncbi.nlm.nih.gov/pubmed/21903084)
Summary: The authors demonstrate a new method for generating textures by constraining a novel noise image to match the layer statistics of a neural network.

Submitted by Assigned_Reviewer_3

In this submission, the authors develop a new texture synthesis algorithm, that begins with the ideas and general methodology of the classical Portilla-Simoncelli texture synthesis algorithm, and augments it with the rich feature set provided by convolutional neural nets. This procedure overcomes a major limitation of Portilla-Simoncelli -- its dependence on a hand-crafted feature set. The convnets, while trained on some other supervised objective (eg classification on imagenet), provide a possible way forward. The authors demonstrate compelling texture synthesis, and some simple and intuitive observations of how generated textures depend on choices of architecture and layer.

For clarity, originality, and significance, this paper gets a thumbs up. For quality, I've given this paper a 6, but I would certainly bump it up higher if the authors convincingly address and fix a few crucial points. In no particular order:

(1) At the moment, the paper is framed in terms of its potential for studying the ventral stream. I can see why this is a tempting proposition, given how Portilla-Simoncelli has fed into to the Freeman et al V2 studies. However, the authors aren't really doing anything in this direction in the body of the paper. The closest they come to perception is in Section 6, wherein they tack a classifier onto the summary statistics collected at a particular layer of VGG. Yet the improvement here with depth seems more to me a statement about the representation that VGG is learning in order to perform well on imagenet, and less about textures and the brain. This could end up a useful tool for studying the internal representations in deep nets, but the connection to the ventral stream is too vague. I think it would be better off if this link was less central, and made into a discussion point, with the texture synthesis in its own right being the focus of the paper.

If the authors wish to propose that the summary statistics collected at different layers of a convnet will be useful for studying hierarchical representations in the brain, then they must acknowledge that these two architectures engage different tradeoffs between scale and complexity. By scale here I mean the size of the receptive field; by complexity, I mean the degree of nonlinearity. In a convnet like VGG, these two factors are correlated: as one goes deeper, units are integrating over a larger spatial region, and have cumulatively computed a more complex nonlinearity. In the ventral stream, this is not the case: as early as V1, the representation is multiscale, including simple cells with lower spatial frequency tuning (i.e. larger RFs). Portilla-Simoncelli is useful as a tool for probing the V1->V2 transformation precisely because it parallels the multiscale feature of the physiology; a convnet is designed with different architectural constraints that may make it less analogous to physiology in this regard.

(2) The authors make a point that the features in Portilla-Simoncelli are hand-crafted, but those proposed in this paper are not. This is mostly true, *except* the choice of the correlation summary statistic *is* hand-crafted. There is no explicit reason for them to choose this, other than it works. For the record, the term "correlation" is misleading here (perhaps in signal processing terms it's a correlation, but it's not normalised, so I would have described it as a covariance).

(3) It's really important to tell us how many (cumulative) summary statistics are being enforced. If I get the numbers right (from inspecting VGG), by pool1 there are 3 * 64*63/2 = 6048 parameters collected, this goes up to 30k by pool2, 194k by pool3, 0.8M by pool4, and 1.5M by pool5. These numbers should go in Figure 2. That's a *lot* of parameters (especially when the number of pixels is ~50k?) and it makes it hard to interpret these as "summary" statistics. The authors should comment on what this means.

(4) I'm at a loss as to why conv5 and/or pool5 are not shown in Figure 2. They're clearly part of VGG, and it's weird that they are omitted without mention.

(5) I'm confused as to why the yellow boxes in the bottom left hand corner of each P-S source texture are showing up in the bottom left hand corner of each synthesised image. If the statistics are stationary, and the seed is white noise, then there's no reason for this feature to be generated precisely here. This looks like a boundary condition artefact. If boundaries are handled in some special way, or ignored, we need to know.

(6) It's pretty topical, so the authors could relate their approach to the recent google inception/deepdream syntheses to make it a more attractive paper. Clearly this work was developed in parallel, and the overall objectives are different. But there is an overlap. For example, my understanding of the deepdream algorithm is that it places pressure on a summary statistic of the current image, namely the variance of one feature map. This is precisely one of the statistics being collected in the algorithm presented here.

(7) As one goes deeper in the net, the texture model is holding on to all the summary statistics computed in previous layers, so the representation is cumulative. This seems like an important point, and yet it is not clearly justified.

** Following author rebuttal: with less emphasis on the biological link, and the authors' proposed amendments, this is an interesting paper. I don't agree with all the interpretations, but it's discussion-provoking and relevant to many communities.
Summary: A great idea. But there's some missing detail, and the biological link is weak.

Submitted by Assigned_Reviewer_4

This paper presents an extension to the Portilla and Simoncelli model of texture generation based on convolutional networks trained on natural scenes. Textures generated by the new convolutional approach appear to match the original image better than the Portill and Simoncelli model, although this has not be quantified. Further, the authors show that linear decoding based on the outputs of the output of the fifth layer of the network reaches the same level of object recognition performance as the very deep VGG network they use. These later results have two apparent inconsistencies. First, the illustration provided in Figure 2 indicates that the texture based model does not encode objects. Second, presumably the full VGG network should still perform better on object recognition tasks.
Summary: This paper presents a modified approach for generating textures based on convolutional very deep networks trained on natural scenes. The results are interesting, although not quantified in all cases. There are also some inconsistencies in the results that need to addressed or discussed.

Submitted by Assigned_Reviewer_5

The paper proposes an interesting approach -- using statistics computed at different layers of a DNN to generate textures, which can then be used to probe representation at different layers of the visual hierarchy.

This paper extends previous methods that used fixed architectures with limited depth to Deep Neural Networks adapted for general image classification takss.

I fully agree with the two specific claims made in the paper: the textures are significantly better than the parameteric texture model of P&S (though they may not match sample-based methods for texture generation); and the stage-wise encoding and synthesis method may provide a useful gradation for testing texture/object representations in the brain.

However, it remains unclear how this model will be used to understand neural visual pathways.

In the spirit of Freeman et al, papers, it may be possible to identify the lowest DNN layer that is metameric for a particular brain area (i.e. neural responses are sensitive to statistics at that level, but not above) -- but what conclusions will this produce?

First, there is already a well accepted hierarchical ordering of visual areas.

Second, representations within DNN are themselves mostly mysterious. The P&S texture model computes specific (and, granted, limited) statistics of the image, and these can be broken down into simple computations, which are mostly quite local, and thus provide theoretical proposals for neural circuits.

Can the same approach be taken with the DNN?

Can this be useful in probing representations of non-textures?

(Not sure how relevant statistical/textural descriptions are for analyzing areas V4, IT.)

Answering these questions is the real challenge of applying DNN techniques to studying the brain.
Summary: The idea of using DNN representation statistics has great merit, but the real challenge will come in applying this method to further theories of visual processing.

The paper would be much stronger if the authors proposed specific questions that can be answered using this model, beyond correlating successive stages of neural processing to successive stages of DNN.

Submitted by Assigned_Reviewer_6

This is an interesting idea but a very limited contribution. This is a twist to a well established method and in order to make this a useful contribution one would need to show an actual application either to better understand complex/deep neural networks or probing into neural selectivity as suggested in the introduction of this paper.
Summary: The paper introduces a new parametric texture model which is essentially a Portilla & Simoncelli type of synthesis model but that uses the output of a state-of-the-art convolutional network instead of steerable filters.

Author Feedback
Author rebuttal: We thank all reviewers for their thorough evaluation of our paper and their insightful comments. The reviewers raise three main criticisms, which we address in detail:

1) Reviewers 1, 2 and 5 find the link to biological vision too weak and not fleshed out. We agree that we have overemphasised the relationship to the ventral visual stream and will limit this point to the discussion in the final version of the paper. While using such textures to study the visual system motivated our work, we stress that the primary contribution of the present paper is the parametric texture model that is able to generate realistic natural textures.

2) Reviewers 4 and 6 question significance and originality or find the contribution limited. However, parametric texture synthesis has been a difficult problem for years. The fact that our solution is an elegant combination of existing concepts does not make it a less important contribution. In fact, the improvement over the state of the art (Portilla & Simoncelli) is so apparent that quantitative analysis is not even necessary. In the light of the numerous experimental studies sparked by the P&S algorithm, it is hard to question the significance of our work for neuroscience. Moreover, the huge interest in Google DeepDreams shows that generating images from discriminatively learned models is also of great interest to the DNN community. Our work was submitted before Google's blog post and offers a much more systematic link to generative modelling than DeepDreams, which simply maximises activations in the network without computing any well-defined summary statistics.

3) Reviewers 2 and 4 point out that the number of parameters in our model is not stated and potentially very large. We apologise for the omission and clarify: For the sake of simplicity we matched the summary statistics in all layers up to layer L. However, in practice we can drop many layers: There is no perceptual difference if we use only layers conv1_1 and pool1-4, in which case we obtain ~177k parameters, which is less than the image (3*256^2 = 197k). We can further reduce the number of parameters with minimal decrease in quality to about 22k by using low-rank approximations to the correlation matrices. Furthermore, matching only the average activation of each feature map in these five layers already produces textures of significantly better quality than P&S while requiring only 1024 parameters (P&S: 1144 for 256x256 images). Therefore we are also confident that our algorithm does not just learn to copy pixel values.

The reviewers raised a few additional points, which we clarify below and in the paper:

4) Reviewer 2: Why is pool5 not shown? Results from layers above pool4 were not shown since they did not show any improvement for the texture synthesis and look like those from 'pool4'.

5) Reviewers 2 and 4: How do the individual layers contribute? As explained above, a subset of layers is already sufficient. However, if summary statistics of low layers (e.g. 'pool1') are not matched the generated textures fail to reproduce low-level image details.

6) Reviewer 2: Why do the yellow boxes reappear at the same position? The VGG network has zero padding on the boundaries of the feature maps in each layer. This leads to some feature maps explicitly encoding the information on the boundaries of the image.

7) Reviewer 4: What about nonlocal correlations? We chose to use only local correlations since it is sufficient to generate high-quality textures from the model. Of course textures from lower layers can be improved using additional constraints (similar to P&S), but since the improvement is negligible for high layers, we preferred the simple approach.

8) Reviewer 3: 'Inconsistency' between results from linear decoding of object identity and texturised version of non-texture image. We believe that what the reviewer calls inconsistency is in fact to be expected and provides a profound insight into how CNNs encode object information. The convolutional representations in the network are shift-equivariant and the network's task (object recognition) agnostic to spatial information, so we expect the dense layers to read out object information independently from the spatial information in the feature maps. We show that this is indeed the case: a linear classifier on the correlation matrix of layer 'pool5' comes close to the performance of the full network (87.7% vs. 88.6% top 5 accuracy). Thus, using our textures we can map out the invariance properties in the network's classification response that arises from the shift invariance of the task: Our texturised versions of non-texture images can be interpreted as a new type of adversarial images that look the same to the network but different to humans. In the revised version of the paper we will explain this link at more detail and show that indeed the original VGG network classifies such textures in the same way as the original source image.